# Diagnostic accuracy of self-collected anterior nasal swabs for SARS-CoV-2 RT-PCR testing

T. Corocher,[1,2] K. Edwards,[1,2] Y. Hersusianto,[1,3] D. Campbell,[4,5] P. Monagle,[6,7,8,9] P. Ho,[1,2,10,11,12] the INHERIT Trial Group

**ABSTRACT**   This prospective sub-study aimed to compare the diagnostic accuracy of the self-collected anterior nasal swab (ANS; Rhinoswab) against that of healthcare professional (HCP)-collected combined throat and nasal swab (CTN). Eligibility criteria for the study included: >5 years of age, a positive severe acute respiratory syndrome coronavirus 2 (SARS-CoV-2) rapid antigen or PCR <72 h prior, and one other participating household member. Samples were tested via reverse transcription polymerase chain reaction (RT-PCR) and included in the analysis where RS and CTN swab were collected on the same day. Indeterminate results were excluded from the analysis. Sensitivity and specificity were used to assess the accuracy of results from self-collected RS. Analysis of the average cycle threshold (Ct) was performed in GraphPad Prism using a paired $t$-test ($P$-value < 0.05, 95% confidence interval [CI]) to assess if there was a significant difference in the Ct values between the HCP-collected CTN swab and the self-collected RS. In total, 583 single-prong result-pairs and 276 double-prong result-pairs were analyzed after exclusion of indeterminate results. The sensitivity of the RS was 84–88.0%, and the specificity was 97–98%. The mean difference in Ct value between the CTN and ANS was ~2 cycles ($P$ < 0.005, ~2.5 CI 95%) with the raw Ct value of discordant results ~10 Ct values higher than concordant results. Furthermore, the sensitivity from days 1 to 10 was 91–96 to 58–69%, while specificity stayed consistent. Overall, our data support the use of the RS as an alternative nasal sampling tool to traditional CTN swabs collected by HCP.

**IMPORTANCE** In this study, we compared the PCR results of self-collected anterior nasal swabs against healthcare professional-collected combined throat and nasal swabs. The sensitivity and specificity of the anterior nasal swab were considered acceptable when compared to World Health Organization recommendations. The sensitivity of the Rhinoswab ANS was higher at the beginning of the trial when compared with the final day of the trial, regardless of the number of prongs tested via PCR. The self-collected Rhinoswab is a valid tool for community screening of severe acute respiratory syndrome coronavirus 2 in vulnerable populations.

**KEYWORDS**   SARS-CoV-2, INHERIT, Intranasal Heparin Trial, nurse-collected, self-collected, CTN, Rhinoswab, Rhinomed

A ccurate, efficient, and readily available community testing is crucial for effective diagnosis of infection and contact tracing during viral outbreaks, as highlighted by the coronavirus disease (COVID-19) pandemic. The reference standard for severe acute respiratory syndrome coronavirus 2 (SARS-CoV-2) detection is reverse transcription polymerase chain reaction (RT-PCR). The World Health Organization (WHO) recommends nasopharyngeal (NP) or combined throat and nose (CTN) sampling using a polyester flocked swab collected by trained healthcare professionals (HCP) (1). A common complaint regarding this sampling approach is the discomfort associated with the procedure (2) and is highlighted in press reports (e.g., 3, 4), which may lead to lower

Address correspondence to T. Corocher, taylor.corocher@nh.org.au.

The authors declare no conflict of interest.

See the funding table on p. 9.

testing compliance within the community. Collection of samples by HCP increases their exposure to infection, a further burden on healthcare services should they fall ill (5). A 2022 Royal Children's Hospital (RCH) study reported that CTN swab sample collection caused children stress, pain, and discomfort during testing for respiratory viruses (6). Testing center locations and wait times may also be a barrier to access for some individuals, who may be more compliant were they able to collect the sample themselves, which illustrates the need for an alternative sampling method that is more user-friendly and easily accessible.

In general, anterior nasal swabs (ANS) have reduced discomfort and were more easily accepted by children; however, they also have a 20% reduced sensitivity when compared to CTN swabs (7–9). The Rhinoswab (RS) (Rhinomed, Melbourne, Australia) is a novel flocked binasal ANS with two prongs, which is produced in both an adult and a smaller child size "Rhinoswab Junior" (RS Jnr). The RS was designed to reduce discomfort during sample collection, especially in children (10, 11), and can be self-collected by following simple instructions. These qualities may make the RS more appealing to patients, thus removing some barriers that may otherwise deter them from seeking testing. The use of swabs that can be self-collected limits the exposure of HCPs to infectious diseases and could also facilitate remote research studies by reducing the number of HCPs required on-site. Unlike traditional ANS, research supports the accuracy of collection with the binasal RS when compared to results from CTN swab (5, 12), and several publications support the diagnostic accuracy of the self-collected method (13–15).

The IntraNasal HEpaRIn Trial (INHERIT; NCT05204550) is currently investigating the efficacy of a heparin nasal spray as prophylaxis against SARS-CoV-2 transmission within households. As part of this clinical trial, samples for SARS-CoV-2 PCR testing were collected by HCP using CTN swabs and self-collected by participants using the RS. The study presented here aimed to assess the diagnostic accuracy (sensitivity and specificity) of the self-collected with the RS compared to traditional HCP-collected CTN swabs.

## MATERIALS AND METHODS

### Study design

The INHERIT study was a double-blind, phase II/III proof of concept cluster randomized placebo-controlled trial of intranasal unfractionated heparin with 1,400 U administered in each nostril three times daily (TDS) for 10 days among household home contacts of index cases for prophylaxis against infection with SARS-CoV-2. The trial population included adults or children in the community setting with confirmed SARS-CoV-2 infection (i.e., a positive result via rapid antigen test or PCR test) and their household contacts. Participants were eligible for inclusion in the INHERIT Trial if they were >5 years of age, had tested positive to SARS-CoV-2 (via a rapid antigen test or PCR test) in the previous 72 h, provided written, informed consent, and with at least one other member of the household who was willing to consent to participation in the study. Exclusion criteria include prior history of heparin allergy, recurrent epistaxis or heparin-induced thrombocytopenia, or if >72 h elapsed since the household index case had tested positive. Participants for the INHERIT Trial were recruited via referrals from a variety of sources, including the Victorian Virtual Emergency Department, Northern Health or its affiliated community healthcare centers, participating general practices, and through radio and social media advertising.

The INHERIT Trial was a prospective study; however, the analysis of diagnostic accuracy between collection methods was not a pre-specified outcome of the trial. This sub-study was performed on samples collected between 1 February 2023 and 31 July 2024. As part of the study design, not all days required collection of both swab types, so only results of sample pairs were included in the analysis (i.e., when both HCP-collected CTN swabs were performed on the same day as the participant self-collected ANS).

## Sample collection methods

Samples were collected by either the reference method: a trained HCP using a polyester flocked CTN swab (FLOQSwab, Copan Diagnostics, Italy) or by the index method: self-collection using an ANS RS (participants over 12 years of age) or RS Jnr (children 5–12 years of age) per manufacturer's instructions (both Rhinomed, Australia). The reference method was selected based on the WHO recommendations for SARS-CoV-2 testing using PCR (1).

All samples for SARS-CoV-2 PCR testing were collected into 3 mL universal transport media (UTM, Copan Diagnostics, Italy) and were split into 1 mL aliquots (without the swab), stored at −80°C within 72 h of collection, and batch-tested weekly as described below. All processing, storage, and testing were performed in the Northern Pathology Victoria laboratory.

A CTN swab was collected on Day 1 (study screening), which was defined as the initial swab collected to determine qualification for inclusion in the INHERIT Trial and prior to commencement of the study drug. Subsequent CTN swabs were collected on days 3, 5, and 10 and placed in 3 mL UTM at the time of collection. The CTN swab was always collected prior to the RS when a nurse was present.

Participants were given instructions on RS self-collection and provided with supplies of the appropriate RS (dependent upon participant age) at Day 1 and were asked to self-collect samples on days 1, 2, 3, 4, 5, and 10 at least 4 h after administering the previous dose of study drug and to store these in the refrigerator until collected by study staff.

Collected RS samples were treated in two ways: (i) for participants recruited between 1 February 2023 and 31 January 2024, the bi-nasal prongs of the RS were separated after collection by the participant, with one prong placed in 3 mL UTM for SARS-CoV-2 testing and the second prong placed in 3 mL sterile distilled water for future testing in an alternative assay; or (ii) for participants recruited between 1 February 2024 and 31 July 2024, both RS prongs were placed into UTM for SARS-CoV-2 testing.

## Test methods: nucleic acid extraction and SARS-CoV-2 PCR

For both HCP-collected CTN swabs and self-collected RS, 300 µL of sample was analyzed. Nucleic acid extraction was performed on the Seegene STARLet liquid handling workstation in combination with the Seegene STARMag Extraction Kit (both In Vitro Diagnostics, Korea). To remove any potential heparin-mediated PCR interference, the extracted nucleic acid was treated with 10 µL of the heparinase I enzyme (0.4 IU/mL; IBEX Technologies, Canada) for 30 min at room temperature as previously described (16). PCR set-up was then performed on the Seegene STARLet with the Seegene Allplex SARS-CoV-2 assay (designed to amplify the SARS-CoV-2 *E*-gene, *N*-gene, and RdRP/*S*-gene) and ran on the C1000 touch thermocycler (Bio-Rad, USA) as per the manufacturer's IFU. The test result was an auto-interpretation by the Seegene Viewer software (In Vitro Diagnostics, Korea) that defines a positive as any combination of two or more of the four genes amplified and an indeterminate result as the *E*-gene only with a Ct value threshold of 40 and a limit of detection of 50 copies/reaction.

## Data selection and analysis

To compare the diagnostic accuracies of the two collection methods, result pairs were identified where both an HCP and an RS were collected on the same date for all INHERIT study samples collected up to 31 July 2024. The results of the CTN swab were considered the reference, or 'true' result, and a result pair was considered concordant if the CTN and RS results matched and discordant if they did not.

Where a result pair contained an indeterminate result, it was excluded from analysis, as in our clinical laboratory these results would not be reported, and recollection of the sample would be recommended. Sensitivity and specificity were used to analyze the data, opposed to positive/negative agreement, as these are the standards the Therapeutic Goods Administration in Australia uses to evaluate diagnostic tools.

Result pairs were further analyzed by separating the data according to whether a single nasal prong or both nasal prongs underwent PCR testing. Sensitivity and specificity were calculated as:

$$\text{Sensitivity} = \frac{\text{true positive}}{(\text{true positive} \, + \, \text{false negative})}$$

$$\text{Specificity} = \frac{\text{true negative}}{(\text{true negative} \, + \, \text{false positive})}$$

A paired $t$-test (GraphPad Prism, Dotmatics, USA) was used to compare concordant positive result pairs, with a $P$-value threshold of <0.05 and 95% confidence interval (CI). Discordant result pairs were excluded from this data set, as a negative result does not return a Ct value. Separately, the mean Ct values of discordant and concordant pairs were calculated to determine if discordant pairs were likely to have higher Ct values on average.

To determine if there was any significant difference in the Ct values within result pairs, a paired $t$-test (GraphPad Prism, Dotmatics, USA) compared concordant positive result pairs (CTN against RS), with a $P$-value threshold of <0.05 and 95% confidence interval (CI). Separately, the average Ct values of discordant and concordant pairs were calculated to determine if discordant pairs were likely to have higher Ct values on average between samples collected with either one or two prongs.

We further analyzed the results by calculating average Ct values and the sensitivity and specificity after separating the data according to the day of swab collection (study day).

To evaluate the diagnostic performance of the RS Jnr, we performed the above analyses on only the result pairs of participants <12 years of age according to participant date of birth at study enrollment. Due to the smaller sample size, this data set was not further grouped by how many nasal prongs were tested as above.

## RESULTS

A total of 930 result pairs were identified as being collected on the same day, with 71 result pairs being excluded due to indeterminate results. As seen in Table 1, a total of 859 result pairs were analyzed, comprising 583 single-prong pairs and 276 two-prong pairs. The percentage of indeterminate samples excluded from analysis was consistent, regardless of the number of prongs tested (~5% for one prong, and ~6% for two prongs) (Table 1).

Overall, the sensitivity was 88 (two-prongs) and 84% (one-prong), and the specificity was 97 (two-prongs) and 98% (one-prong) (Table 1).

To assess the accuracy of the self-collected RS compared to the HCP-collected CTN swab, a paired $t$-test was performed. The average difference in Ct values between the RS with two prongs and the HCP CTN swab types was statistically significant at around 2 Ct values for each gene (all $P$-values were <0.0001) (Table 2). Similar results were seen where only one prong was tested (Table 2). A paired $t$-test analysis compared the average difference in Ct values (CTN–RS) to assess for statistical significance between swab types. The average difference in Ct values was ~2 cycles with a $P$-value of <0.05, inferring a statistically significant increase in the Ct values of the self-collected RS (2.027–2.405, gene-dependent) with a 95% CI of 1.426–2.955 (Table 2). The "n" of each gene varies, as the PCR used does not require every gene to be present to determine a SARS-CoV-2 infection.

After assessment of the significant mean difference of Ct values, the mean of the raw Ct values of all positive concordant and discordant pairs was calculated to see if discordant results had a higher Ct value on average compared to concordant pairs (Table 3). The analysis demonstrated that discordant Ct values were 10 Ct values higher (~35 Ct)

**TABLE 1** Initial data analysis of paired data[a]

| | Two prongs | One prong | Total |
|---|---|---|---|
| Pairs collected on the same day (n) | 307 | 623 | 930 |
| Pairs excluded due to indeterminate results (n) | 31 | 40 | 71 |
| Total pairs used in data analysis (n) | 276 | 583 | 859 |
| Total concordant pairs (n) | 254 | 534 | |
| True positive pairs (n) | 129 | 215 | |
| True negative pairs (n) | 125 | 319 | |
| Total discordant pairs (n) | 22 | 49 | |
| False negative (n) | 18 | 41 | |
| False positive (n) | 7 | 8 | |
| Sensitivity (%) | 88% | 84% | |
| Specificity (%) | 97% | 98% | |

[a]Result pairs for the self-collected RS and HCP-collected CTN swabs were identified, and samples with indeterminate results were excluded from the analysis. The number of true positive/true negative and false negative/false positive pairs was determined and the sensitivity/specificity calculated.

on average compared to the concordant Ct values (~25 Ct), regardless of the number of prongs tested (Table 3).

The RS Jnr is a smaller version of the RS used for adults, and given the difference in physical size, a sub-analysis was performed to determine the accuracy of RS Jnr (Table S1) at https://doi.org/10.5281/zenodo.17566954. We found the RS Jnr to have similar sensitivity and specificity (87 and 100%, respectively) to the adult RS. Because of the small sample size (n = 75), we did not further separate the results into one prong vs. two prongs.

When assessing the sensitivity and specificity over time, there was a decrease between study Day 1 and Study Day 10 (Table 4), regardless of the number of nasal prongs tested. However, the sensitivity on study Day 1 (91–96%) was higher than day 10 (58–69%). When only Day 1 samples were used to calculate sensitivity (Table 4), the sensitivity was higher than the combined analysis of all study days displayed in Table 1 (84–88%). Furthermore, the mean raw Ct value of all three genes increased as the study day increased from 24.8 on Day 1 to 32.5 on Day 10 (Table 4).

## DISCUSSION

The aim of our work was to assess the suitability of the self-collected RS as an alternative nasopharyngeal collection method to traditional HCP-collected CTN swabs. The

**TABLE 2** Analysis of the mean difference in Ct values between HCP-collected CTN swabs and self-collected RS[a]

| Two prongs | | | |
|---|---|---|---|
| Target gene | E-gene (n = 133) | N-gene (n = 121) | RdRP/S-gene (n = 129) |
| Difference between CTN and RS Ct values (mean, 95% CI) | 2.052 (1.426–2.678) | 2.027 (1.347–2.706) | 2.228 (1.591–2.865) |
| P-value | <0.0001 | <0.0001 | <0.0001 |
| One prong | | | |
| Target gene | E-gene (n = 213) | N-gene (n = 206) | RdRP/S-gene (n = 203) |
| Difference between CTN and RS Ct values (mean, 95% CI) | 2.283 (1.763–2.803) | 2.355 (1.829–2.882) | 2.405 (1.855–2.955) |
| P-value | <0.0001 | <0.0001 | <0.0001 |

[a]The mean difference in Ct values for each paired self-collected RS and HCP-collected CTN swabs was assessed using a paired t-test. The mean Ct value is presented alongside the 95% CI and P-value < 0.05 being considered significant.

**TABLE 3** Average Ct values of concordant and discordant result pairs[a]

| Two prongs | | | |
|---|---|---|---|
| **Gene** | *E*-gene | *N*-gene | *RdRP/S*-gene |
| Discordant | 35 | 35.6 | 36.3 |
| Concordant | 25.9 | 28.5 | 27.4 |
| One prong | | | |
| **Gene** | *E*-gene | *N*-gene | *RdRP/S*-gene |
| Discordant | 33.3 | 33.0 | 34 |
| Concordant | 26.6 | 26.8 | 27.4 |

[a]The average Ct value of positive results was calculated for each gene tested for either two prongs or one prong.

sensitivity and specificity of the RS were calculated with the HCP-collected CTN as the reference. Furthermore, a sub-analysis of the impact of the number of nostrils sampled was also investigated. For RS sampling of two prongs, there was a total of 276 result pairs analyzed, and for the RS sampling of a single nostril, 583 result pairs were analyzed (Table 1).

The sensitivity of the RS with two prongs was 84%, while that with one prong was 88%, and the specificity for both was 97 and 98%, respectively (Table 1). The RS meets the TGA requirement for sensitivity of diagnostic assays (80%) but is borderline for the specificity requirement (98%). The slight reduction in specificity of the two-prong samples (97%) may be due to the lower sample number when compared to the single prong (98%) (17, 18). However, it is important to note that the overall sensitivity of the RS for both single- and double-prong groups was lower (Table 1) than the previously published data testing only the RS Jnr (91%; [19–21]) but higher than published data regarding the adult RS (80.7%; [22]).

Subsequently, the mean difference in Ct values between collection methods was calculated, and statistical significance was determined using a paired *t*-test. The mean difference in Ct values between swab types was ~2 Ct values for all genes amplified, and the difference was considered statistically significant (*P*-value < 0.001) (Table 2). As the cut-off threshold for the PCR is a Ct of 40, a difference of 2 Ct values would only change the qualitative result (e.g., negative/positive) where the sample Ct value is >38. Furthermore, as SARS-CoV-2 diagnostic tests do not have a standard curve and are not

**TABLE 4** Analysis of sensitivity, specificity, and average Ct value of the RS by study day[a]

| Two prongs | | |
|---|---|---|
| **Study day** | **Sensitivity (%)** | **Specificity (%)** |
| 1 | 96% | 97% |
| 3 | 89% | 100% |
| 5 | 83% | 89% |
| 10 | 69% | 100% |
| One prong | | |
| **Study day** | **Sensitivity (%)** | **Specificity (%)** |
| 1 | 91% | 97% |
| 3 | 86% | 98% |
| 5 | 88% | 98% |
| 10 | 58% | 98% |
| Average Ct value | | |
| **Study day** | **Two prongs** | **One prong** |
| 1 | 24.8 | 25.9 |
| 3 | 27.9 | 28.0 |
| 5 | 30.8 | 29.1 |
| 10 | 32.5 | 31.8 |

[a]Data were subdivided by study day (1, 3, 5, 10), and the sensitivity and specificity for samples testing two prongs and one prong were calculated. Furthermore, the average Ct value of positive results was determined for each study day.

quantitative, the Ct value cannot be equated with viral load. A difference in Ct values does not change the clinical diagnosis unless the result borders the cycle threshold.

Analysis of the differences in mean raw Ct values of discordant samples was calculated and found to be ~10 Ct values higher than concordant samples (Table 3), suggesting discordant results occur more often when Ct values are higher (5, 19). The Ct values >38 would be affected. In our study, Ct values >38 in the *N*-gene were seen in only 2.6% of the one-prong samples and 3.7% of two-prong samples. Therefore, such a difference is unlikely to be of clinical significance, as SARS-CoV-2 diagnostic tests are not quantitative, and the result is mostly interpreted as does the patient have SARS-CoV-2 or not. Without the use of a standard curve to enable accurate quantification of viral load, the individual Ct values infer little clinical significance. A previous publication (19) considered a swab as non-inferior if the 95% CI was <3; thus, we conclude that the RS is non-inferior to the CTN as the mean difference in Ct values of ~2 cycles, and the 95% CI < 3 RS.

The RS Jnr was used when appropriate for participants ≤12 years. The sensitivity and specificity of samples collected with the RS Jnr were consistent with the results of the overall RS data analysis (Table S1) at https://doi.org/10.5281/zenodo.17566954, encouraging the use of RS for collection in children, especially as reports show a decreased discomfort and increased willingness to test using the RS Jnr compared to traditional flocked swab in children (5, 7).

The sensitivity of the RS was 91–96% on recruitment of participants and 58–69% by Day 10 (Table 4). For the INHERIT Trial, the study day does not necessarily equal infection day, as study inclusion requires the household to be recruited within 72 h of the index case testing positive. As the average Ct value of positive SARS-CoV-2 results on Study Day 10 was >30 (Table 4), it is reasonable to assume that the decreasing sensitivity was likely due to the stage of viral infection of participants. Supporting this are recent publications that demonstrate an increase in false negatives with increasing time since symptom onset (20, 21). Additionally, any variation in sample collection, such as not leaving the RS in the nasal passage for the recommended 60 s, would be likely to impact results. Some argue that test frequency and turnaround time are more of a priority in epidemic screening compared to the sensitivity of the test (23), especially as SARS-CoV-2 has yet to be cultured 9 days after the onset of symptoms (24). As the RS allows for self-collection, it is suitable for use in geographically remote areas and additionally lessens the exposure to healthcare workers. It may be a valuable alternative collection method during the initial infectious stages for assessment of large populations during pandemics.

As results between single- or double-prong samples were consistent, we suggest that the RS would also be useful during research that requires specific media for sample collection that may be incompatible with other test methods. For instance, UTM has autofluorescence and interferes with any assays that measure fluorescence; one prong could be placed in UTM and the other in distilled water for other testing. This would allow for the collection of a single sample per timepoint, rather than multiple, leading to a lower burden on trial participants. It was also common to pool samples for testing during the COVID-19 pandemic to decrease the time to result. Any positive pools were then retested as individual samples, potentially delaying the time to result and/or requiring sample recollection if the original swab had been stored for longer than recommended by the manufacturer (17, 18). In this situation, the use of the RS for sample collection would eliminate the need for sample re-collection, as one prong could be used for pooled sample testing, with the second prong retained for individual retesting if required. However, individual laboratories would need to assess if the small decrease in sensitivity of testing one prong would be an acceptable trade-off for the decreased time to results. In addition, the availability of alternative swab types, such as the RS, would increase options for healthcare providers in the context of supply shortages like those seen during the pandemic.

## Limitations

As the INHERIT Trial currently remains blinded, it is unknown which samples may have been affected by heparin-mediated PCR interference, which despite treatment with heparinase I, would result in slightly elevated Ct values. Our previously published work noted that heparinase I had difficulty in recovering samples with Ct values >35, which had heparinase I treatment (16), which may be a compounding factor contributing to the sensitivity of the RS. In addition, as the RS has a greater surface area and is exposed to the nasal passages for an extended period of time compared to the traditional flocked CTN, it may absorb more heparin, increasing the degree of interference.

Another limitation of this study was that RS self-collection was only supervised by study staff on Day 1; hence, we relied on compliance from the study participants to follow the instructions provided. The HCP-collected CTN swab was collected prior to the RS when a nurse was present, which may result in some bias within the results as regards the quality of the second swab collected. Furthermore, false negative results may reflect a low amount of cell material, which is determined by the simultaneous detection of an expressed human gene and was not controlled for in this study (25). It is difficult to address the populations that would have a greater benefit from the RS, as the demographic data were not assessed. Finally, there was a small number of <5 yo recruited for the study, leading to caution when considering the results of this age bracket compared to previously published data.

## Conclusions

Based on our analyses, we find that the self-collected RS is non-inferior to the HCP-collected CTN swab and would be a useful tool for community-based infection surveillance, increasing access to testing in remote regions and to decrease exposure to HCPs (23, 26). The ability to separate the RS prongs into two samples makes it ideal for use situations where multiple tests are required from the one specimen. Current reports in the literature indicate that both the RS and ANS sampling methods were rated as more comfortable and preferred by users over the CTN sampling method, which could impact testing compliance vital for containment of infectious outbreaks (5, 9, 10, 18).

## ACKNOWLEDGMENTS

Trial Steering Committee: Gary P. Anderson, Andrew Davidson, Anneke Grobler, Damian Purcell, Craig Aboltins, Shidan Tosif, Steven Y.C. Tong, Kerry Fitzmaurice, Michelle McIntosh, Timothy Hinks, Barry Dixon, Marion Angelika Kainer, Paul Monagle, and Donald Campbell. Operations Team: Paul Monagle, Donald Campbell, Philip Vukovic, Vasiliki Karlaftis, Jodi Hislop, Natasha Letunica, Tanya Anand, Abeerah Sadiq, Rizelle Santos, Amelia Maria Sfameni, Madonna Sanchez, Derwin Or, Seyma Mulayim, Jan Trisha Sanchez, Chantal Attard, Vera Ignjatovic, Sharon Tay, Christopher Dong, Avanthika Lakshmanan, and Valerie Yee. Lab Team: Prahlad Ho, Taylor Corocher, Kira Edwards, Yvonne Hersusianto, and Hui Yin Lim. Pharmacy Team: Vinod Chellaram, Nivein Yenis, Seraphina Kwak, Jier Nguyen, Elise Simonato, and Blazhenka-Angelika Gjorgjevska. Long-COVID Team: Sarosh Sadiq, Sophie May Roberts, Yuqi Mao, and Tissa Wijeratne. Safety Review Committee: Fiona Newall, Meredith Wiggins, Donald Campbell, Gary P. Anderson, Paul Monagle, and Shidan Tosif. Data Safety Monitoring Committee: Mark Crowther, Thomas Sullivan, Noel Cranswick, Huyen Tran, Joe Toressi, and Susan Russel. Sub-Investigators: Loren Sher, Alexandra Brown, Hui Bing Ooi, Sue Lynn Lim, Jessica Josephine Edwards, Lucy Victoria Hanlon, Suguna Ganesan, Jingnan Lisa Zhang, Laura Fangqi Chen, Kit Yee Tang, Henry Wong, Matthew Alexander Murphy, Jesica Oktaviana, and Gopika Krishnamurthy.

T. Corocher: methodology, data curation, formal analysis, investigation, writing—original draft, writing—review and editing, and visualization. K. Edwards: conceptualization, methodology, data curation, formal analysis, investigation, writing—original draft, writing—review and editing, visualization, and project administration. Y. Hersusianto:

writing—review and editing and supervision. D. Campbell: conceptualization, writing—review and editing, supervision, and funding Acquisition. P. Monagle: conceptualization, writing—review and editing, supervision, and funding acquisition. P. Ho: conceptualization, writing—review and editing and supervision. All subjects were recruited into the INHERIT randomized controlled trial, and the specific roles of the clinical trial team are stated in Appendix 1.

This work was funded by a grant from the Victorian State Government Department of Jobs, Skills, Industry and Regions (DJSIR). Rhinoswabs were kindly supplied by Rhinomed Australia. Neither funding source had any role in the study design, conduct of the research, collection, analysis or interpretation of data, in the writing of the report, or the decision to submit the article for publication.

## AUTHOR AFFILIATIONS

[1]Northern Clinical Diagnostics & Thrombovascular Research (NECTAR) Centre, Northern Health, Melbourne, Victoria, Australia
[2]Northern Pathology Victoria, Northern Health, Melbourne, Victoria, Australia
[3]Infectious Diseases, Northern Health, Melbourne, Victoria, Australia
[4]Hospitals without walls, Northern Health, Melbourne, Victoria, Australia
[5]Department of Medicine – Southern Clinical School, Monash University, Clayton, Victoria, Australia
[6]Department of Paediatrics, University of Melbourne, Melbourne, Victoria, Australia
[7]Murdoch Children's Research Institute, Melbourne, Victoria, Australia
[8]Department of Haematology, Royal Children's Hospital, Parkville, Victoria, Australia
[9]Kids Cancer Centre, Sydney Children's Hospital, Randwick, New South Wales, Australia
[10]Department of Haematology, Northern Health, Melbourne, Victoria, Australia
[11]Department of Medicine – Northern Health, University of Melbourne, Melbourne, Victoria, Australia
[12]Australian Centre for Blood Diseases, Monash University, Clayton, Victoria, Australia

## AUTHOR ORCIDs

T. Corocher http://orcid.org/0000-0003-1127-0500
K. Edwards http://orcid.org/0000-0002-8169-5187
D. Campbell http://orcid.org/0009-0006-2639-7001
P. Monagle http://orcid.org/0000-0002-3970-8984
P. Ho http://orcid.org/0000-0003-1875-3927

## FUNDING

| Funder | Grant(s) | Author(s) |
| --- | --- | --- |
| Victoria State Government Department of Jobs, Skills, Industry and Regions | | Donald Campbell |
| Victoria State Government Department of Jobs, Skills, Industry and Regions | | Paul Monagle |

## AUTHOR CONTRIBUTIONS

T. Corocher, Data curation, Formal analysis, Investigation, Methodology, Visualization | K. Edwards, Conceptualization, Data curation, Formal analysis, Methodology, Project administration | Y. Hersusianto, Supervision | D. Campbell, Conceptualization, Supervision | P. Monagle, Conceptualization, Supervision | P. Ho, Conceptualization, Supervision.

## DATA AVAILABILITY

The data analysed for this work were openly available in Github at https://doi.org/10.5281/zenodo.17566954. Table S1 is also located in this repository.

## ETHICS APPROVAL

The INHERIT study was prospectively registered (NCT05204550), and ethical approval for the conduct of the trial was granted by St. Vincent's Hospital (Melbourne) Human Research Ethics Committee (project number: 83609). Site-specific governance approval was granted by the Northern Health Research Office, and all participants (or their legal guardians) provided written informed consent.

## ADDITIONAL FILES

The following material is available online.

### Open Peer Review

**PEER REVIEW HISTORY (review-history.pdf).** An accounting of the reviewer comments and feedback.

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
