## [Reviewer comments · Microbiology Spectrum]

Microbiology Spectrum

Diagnostic accuracy of self-collected anterior nasal swabs for SARS-CoV-2 RT-qPCR testing

T. Corocher, K. Edwards, Y. Hersusianto, D. Campbell, P. Monagle, P. Ho, and the INHERIT Trial Group

Corresponding Author(s): Taylor Corocher, Northern Health

Review Timeline:

Submission Date:	June 9, 2025
Editorial Decision:	July 7, 2025
Revision Received:	October 22, 2025
Accepted:	October 27, 2025

Editor: Wendy Szymczak

Reviewer(s): The reviewers have opted to remain anonymous.

Transaction Report:

DOI: <https://doi.org/10.1128/spectrum.01711-25>

Re: Spectrum01711-25 (Diagnostic accuracy of self-collected anterior nasal swabs for SARS-CoV-2 rt-qPCR testing)

Dear Dr. Taylor Corocher:

Thank you for the privilege of reviewing your work. Below you will find my comments, instructions from the Spectrum editorial office, and the reviewer comments.

Revision Guidelines

Sincerely,
Wendy Szymczak
Editor
Microbiology Spectrum

Reviewer #1 (Comments for the Author):

Thank you for the opportunity to help review the manuscript entitled, "Diagnostic accuracy of self-collected anterior nasal swabs for SARS-CoV-2 rt-qPCR testing. Corocher et al., investigated the use of a self-collected nasal swab sample for SARS-CoV-2 PCR as compared to healthcare professional collected swabs (combined nasal and throat). The study provides promising evidence in support of using self-collected rhinoswabs as an alternative to provider-collected CTN swabs for SARS-CoV-2

detection. I have a few comments and questions that would help clarify the manuscript.

Line 73-75 Please define the acronyms used (ANS, RS, CTN) in the abstract

Line 92-104 children are cited as being the ones that would benefit from anterior nasal swab testing but children <5 were excluded from the study and there was more limited data collected with the RS jr. Future studies would be necessary to show that RS would have similar accuracy in young children. Additionally, lack of demographic details makes it hard to assess applicability to specific populations.

Line 99-100 The study did not report on participant comfort or preference for the Rhinoswab vs the CTN swab. Even though other studies have shown a preference for nasal swabs, acceptability of the RS would be important for widespread use.

Line 135-145 Were samples stored, processed, run, etc., in identical conditions, in the same laboratory?

Line 162-164 How were the samples stored until they were tested?

Line 166-171 The study did not mention random assignment of collection order or method which could introduce bias (like if one swab type was always collected first).

The sample exclusion seems really high. Please clarify the reasons for exclusion beyond same day collection and indeterminate results.

Along the same line, excluding the indeterminate samples, especially if they occurred more frequently in one swab type, could overestimate the true accuracy.

The presentation of the data would benefit from clearer and more standardized tables. Also, the data should have more labels. For example, in places where Ct values are reported, that should be listed as the unit (Table 4 and 5)

Table 5 There may be a better way to display these data with more clarity. For example, what is a positive discordant pair? Is that a false positive? So 4 samples were negative by CTN and positive by NS? Why does the E gene target have an N=32? Additionally, the results of this table should be described in the main text.

Discussion: some of the information in this section may be better suited in the results section or should at least be described in the results first.

Line 255- 261: Could the lower viral load have to do with the heparin in the sense that maybe it had some activity as a prophylaxis? The authors addressed the possible inhibitory effect of the heparin itself

Line 262-267: While the ~2 Ct difference is statistically significant, the claim that it "rarely changes the qualitative result" needs supporting data (e.g., how many samples had Ct values near the assay's positivity threshold where a 2-cycle shift would matter?). An important note about comparing Ct values, is that this study did not correlate these values with clinical outcomes or infectiousness, so this limits the interpretation of what a 2 cycle difference may be for public health decisions.

Reviewer #2 (Comments for the Author):

The authors present a specimen collection performance comparison between self-collected anterior nasal swabs versus provider-collected combined throat and nasal swabs for the detection of SARS-CoV-2 as a sub-study in an ongoing clinical trial. This collection method seems promising and useful due to the ease of self-collection, particularly in children. The manuscript could use improvement with regards to data presentation, conclusions drawn, and clarity of methods. See attachment for detailed items to address.

Reviewer #3 (Comments for the Author):

Thank you for the review opportunity

In this manuscript, the authors evaluated the performance of self-collected anterior nasal swabs for detecting SARS-CoV-2. The data demonstrated that the sensitivity and specificity of the self-collected swabs were comparable to those collected by health-care providers, suggesting that self-collected swabs are useful tools for community screening of SARS-CoV-2. The study is well designed and the data support authors' conclusion. However, the manuscript would benefit from further revision and clarification. Here are my specific comments

1. Some tables and figures appear unnecessary for the limited amount of data, which makes the manuscript hard to follow. Please consider significant shortening/streamlining the manuscript. For example, removing figure 1 and table 3 may improve clarity.

2. Line 193: can you provide more details about "indeterminate results"? What caused these results? Were they due to failed internal control or failed RNase P assay? additionally, please clarify whether these indeterminate results occurred in self-collected swabs, health-care provider collected swabs or both. This information is important especially if RNase P assay was

used to assess sample quality.

3. Are there significant difference between 2 Prongs arm or 1 prong arm? Please consider presenting these groups separately in tables 1 and 2.

4. Please consider reformatting table 2.the layout of table 2 makes it difficult to follow.

5. Table 5: a 2 by 2 table may present the data more clearly. Please consider revising.

6. Line 142: please clarify the time interval between self-collection and the time when the swabs were picked up by the study staff. Prolonged storage of swabs in the refrigerator may affect the integrity of viral RNA and impact results.

7. Line 174: Ct values should be interpreted with caution as they are only a surrogate measure of viral load in a sample. Other factors also influence the Ct values. Please re-phrase this sentence.

8. There are grammatical errors throughout the manuscript. please revise

The authors present a specimen collection performance comparison between self-collected anterior nasal swabs versus provider-collected combined throat and nasal swabs for the detection of SARS-CoV-2 as a sub-study in an ongoing clinical trial. This collection method seems promising and useful due to the ease of self-collection, particularly in children. The manuscript could use improvement with regards to data presentation, conclusions drawn, and clarity of methods.

Major points

- Lines 94-95 (nasal swabs are less sensitive than CTN swabs) and 102-103 (nasal swabs are as accurate as CTN swabs) seem to be in direct conflict.
- **Table 1.** Instead of presenting the comparison and calculating sensitivity/specificity, the data should be presented as percent agreement between CTN and the 2 prong swab, and then a second table with percent agreement between CTN and the 1 prong swab. These could be presented more simply as a 2 x 2 standard table which is more standard and easily interpreted by readers.
- **Methods could be clarified.**
 - Collection methods: How were participants asked to self-collect (were they given instructions on how many seconds/minutes they needed to spend actively sampling, did they have to rotate the swabs in their nostrils, how far up were they told to insert the devices, etc.)?
 - For the two-pronged rhinoswabs, were the prongs separated and placed into UTM vs distilled water by the study participants? Or by study staff?
 - Line 65: When only one prong was in UTM what was going on with the 2nd swab? Was it kept dry or in saline?
 - Assay: What sample input volume was used for the extraction? How were specimens from the 2 pronged samples combined prior to extraction? What is the LOD of the assay? What is the Ct value cutoff for positive/negative? What internal control was used (RNase P)? How many of the gene targets needed to be detected to be considered positive?
 - Define what an indeterminate result is (This could be interpreted a number of ways otherwise. Internal control failure in a negative result? One gene target positive only?)
- **Table 3 and 4.** I don't think it's very informative to do a sub-analysis of test agreement this way. It's expected that viral load will decrease over time from symptom onset in these participants and that test performance will be less reliable with a lower burden of infection. It might be more helpful to show a scatterplot with Ct value correlations between paired RS and CTN samples to demonstrate that both methods correlate across a wide range of viral loads.
- **Discussion**

- Line 249: Does UTM actually interfere with RT-PCR signal detection? I would think that extraction/elution buffers would take care of that.
- Lines 253: I don't follow how RS would decrease the number of samples needed for pooling. At best it might make sample storage/tracking simpler without needing to aliquot.
- Another effective discussion point would be including RS as a viable option in the context of supply chain shortages.
- Line 296: Were there criteria for rejecting/excluding samples that were negative for the internal control?

Minor points

- I would refer to testing as RT-PCR, not RT-qPCR since a qualitative (positive/negative) rather than a quantitative result is ultimately reported.
- I would be more careful about the use of the term "screening" throughout the text: this study did not address the question of routine testing of populations without symptoms.

Reviewer 1

1. Line 73-75 Please define the acronyms used (ANS, RS, CTN) in the abstract

The acronyms have been described in the abstract, lines 56-59.

2. Line 92-104 children are cited as being the ones that would benefit from anterior nasal swab testing but children <5 were excluded from the study and there was more limited data collected with the RS jr. Future studies would be necessary to show that RS would have similar accuracy in young children. Additionally, lack of demographic details makes it hard to assess applicability to specific populations.

We have included a paragraph on the benefits to patients other than children who are reluctant, or those who are unable to access a testing centre easily on lines 93-9 and further have addressed the point about demographic details and limited testing of <5yo in the limitation section of the paper in lines 355-358.

3. Line 99-100 The study did not report on participant comfort or preference for the Rhinoswab vs the CTN swab. Even though other studies have shown a preference for nasal swabs, acceptability of the RS would be important for widespread use.

We acknowledge that there is a no data regarding comfort of preference of the RS swab in this trial, a sentence has been added at line 112-114 to highlight that this trial focuses on the accuracy of the swab and that more studies must be performed to demonstrate acceptability to users of the RS swab.

4. Line 135-145 Were samples stored, processed, run, etc., in identical conditions, in the same laboratory?

A sentence on the processing, storage and analysis of the samples has been included at lines 153-158.

5. Line 162-164 How were the samples stored until they were tested?

Participants were instructed to store samples in the refrigerator until collected by study staff. Samples were stored at -80C within 72 hours of collection after being split into 1 mL aliquots. The storage conditions of the samples are provided in lines 149-152.

6. Line 166-171 The study did not mention random assignment of collection order or method which could introduce bias (like if one swab type was always collected first).

The nurse collected swab was always collected first at the 'screening' visit, and the participant received instructions on self-collection with the RS. However, on subsequent visits the self-collected RS was performed in the hours prior to the nurse visit. Lines 157-158 have been adjusted.

7. The sample exclusion seems really high. Please clarify the reasons for exclusion beyond same day collection and indeterminate results.

The only exclusions for samples were the samples being collected on the same day, and the results being either positive or negative. There was no other exclusion. The total sample exclusion was 71 result pairs out of 930 (~7%).

Northern Health

Along the same line, excluding the indeterminate samples, especially if they occurred more frequently in one swab type, could overestimate the true accuracy.

We have added a point to reflect this in the manuscript on lines 351-352.

8. The presentation of the data would benefit from clearer and more standardized tables.

We have standardised the data presented in the tables as recommended.

9. Also, the data should have more labels. For example, in places where Ct values are reported, that should be listed as the unit (Table 4 and 5)

The appropriate units have been added to the tables.

10. Table 5 There may be a better way to display these data with more clarity. For example, what is a positive discordant pair? Is that a false positive? So 4 samples were negative by CTN and positive by NS? Why does the E gene target have an N=32? Additionally, the results of this table should be described in the main text.

The table has been adjusted to reflect pairs where the results were a false positive or false negative. The difference in the number of results assessed for the E-gene is because the E-gene is often present with either both the N-gene and S-gene or just one of them.

A paragraph has been added at line 230-231 to explain the figure in more detail.

11. Discussion: some of the information in this section may be better suited in the results section or should at least be described in the results first.

The discussion has been edited to ensure information suitable is in the section.

12. Line 255- 261: Could the lower viral load have to do with the heparin in the sense that maybe it had some activity as a prophylaxis? The authors addressed the possible inhibitory effect of the heparin itself

At this time, we are unable to assume the Ct values closer to the threshold (inferring a lower viral load in the collected sample) is due to heparin as the samples are blinded and this information is not accessible. However, as households were randomly assigned to heparin or placebo on a 1:1 basis, any effect would be equally distributed.

13. Line 262-267: While the ~2 Ct difference is statistically significant, the claim that it "rarely changes the qualitative result" needs supporting data (e.g., how many samples had Ct values near the assay's positivity threshold where a 2-cycle shift would matter?). An important note about comparing Ct values, is that this study did not correlate these values with clinical outcomes or infectiousness, so this limits the interpretation of what a 2 cycle difference may be for public health decisions.

The qualitative result of a SARs-CoV-2 PCR would change only if the Ct value was within the range of variance from the threshold (which is 40 in this case). As our variance is ~ 2 cycles, we would assume that the qualitative result, if they do or not have COVID-19, would change from only if the Ct value was 38 or above. The text (lines 295-301) has been changed to reflect the uncertainty of this statement in relation to the lack of clinical information and in the absence of a standard curve to calculate the viral load. The impact of the lack of clinical information has been acknowledged in the text.

Reviewer 2

1. Lines 94-95 (nasal swabs are less sensitive than CTN swabs) and 102-103 (nasal swabs are as accurate as CTN swabs) seem to be in direct conflict.

The reduction of sensitivity cited was reported from other brands of anterior nasal swabs when compared to traditional CTN swabs, rather than the Rhinoswab specifically. The studies cited report accuracy of the binasal Rhinoswab itself and are followed by citations supporting the act of self-collection versus collection by a health care professional. The text has been adjusted to clarify this on lines 97 and 98.

2. **Table 1.** Instead of presenting the comparison and calculating sensitivity/specificity, the data should be presented as percent agreement between CTN and the 2 prong swab, and then a second table with percent agreement between CTN and the 1 prong swab. These could be presented more simply as a 2 x 2 standard table which is more standard and easily interpreted by readers.

The reviewer's suggestion is perfectly reasonable, and we had originally debated presenting the data as percent positive/negative agreement versus sensitivity/specificity. However, as most diagnostic papers presented sensitivity/specificity, and the Australian drug regulatory agency (TGA) guidelines require minimum sensitivity/specificity for regulatory approval, we decided to follow these methods of analyses.

3. Collection methods: How were participants asked to self-collect (were they given instructions on how many seconds/minutes they needed to spend actively sampling, did they have to rotate the swabs in their nostrils, how far up were they told to insert the devices, etc.)?

Clarification of the methods has been added at lines 149-158.

4. For the two-pronged rhinoswabs, were the prongs separated and placed into UTM vs distilled water by the study participants? Or by study staff?

We have specified that the prongs were separated by the participants in line 154.

5. Line 65: When only one prong was in UTM what was going on with the 2nd swab? Was it kept dry or in saline?

We specify in line 155-156 of the manuscript that the second prong was placed in 3 mL of sterile distilled water and are planned for future testing using a method for which UTM causes interference.

6. Assay: What sample input volume was used for the extraction? How were specimens from the 2 pronged samples combined prior to extraction? What is the LOD of the assay? What is the Ct value cutoff for positive/negative? What internal control was used (RNase P)? How many of the gene targets needed to be detected to be considered positive?

The following have been added to address each point in this comment:

- Added a sentence at line 141-144 to describe that the 1 mL aliquots did not have the swabs present
- Added a sentence at line 160 to state that 300 ul was used as the extraction volume
- Added a sentence at 167-170 defining the LoD as 50 copies per reaction and the definition of the Ct value cutoff
- Added a sentence defining the Ct value cutoff

7. Define what an indeterminate result is (This could be interpreted a number of ways otherwise. Internal control failure in a negative result? One gene target positive only?)

The definition of an indeterminate result is defined on line 175.

8. **Table 3 and 4.** I don't think it's very informative to do a sub-analysis of test agreement this way. It's expected that viral load will decrease over time from symptom onset in these participants and that test performance will be less reliable with a lower burden of infection. It might be more helpful to show a scatterplot with Ct value correlations between paired RS and CTN samples to demonstrate that both methods correlate across a wide range of viral loads.

We have moved the table to the supplementary and refer to it in the results section instead. However, a scatter plot of the RS and CTN paired samples has too many plots to determine how close individual pairs are. Instead, we have simplified and standardised the table to show the sensitivity/specificity over time.

9. Line 249: Does UTM actually interfere with RT-PCR signal detection? I would think that extraction/elution buffers would take care of that.

No, UTM does not interfere with signal detection in PCR assay. We have specified in lines 330-334 that we are referring to other methods, that are not PCR based, such as a fluorescence-based assays.

10. Lines 253: I don't follow how RS would decrease the number of samples needed for pooling. At best it might make sample storage/tracking simpler without needing to aliquot.

Our thinking was that instead of being required to collect a new swab from each individual when a pool tests positive, the first prong is used for pooled testing, and the second for individual testing when required. This section has been reviewed for clarification.

11. Another effective discussion point would be including RS as a viable option in the context of supply chain shortages.

Thank you for the suggestions, the text has been adjusted to include this point at lines 340-342.

12. Line 296: Were there criteria for rejecting/excluding samples that were negative for the internal control?

Results with a negative internal control were considered invalid, and retested until the internal control was within 2 Ct values of the average Ct value of 25.

13. I would refer to testing as RT-PCR, not RT-qPCR since a qualitative (positive/negative) rather than a quantitative result is ultimately reported.

We agree with this point and have amended the article as described.

14. I would be more careful about the use of the term "screening" throughout the text: this study did not address the question of routine testing of populations without symptoms.

Yes, in this situation, screening is referring to initial PCRs from the households, as a positive PCR result was required to be eligible for participation in the study. We have defined this in the manuscript on line 145.

Reviewer 3

1. Some tables and figures appear unnecessary for the limited amount of data, which makes the manuscript hard to follow. Please consider significant shortening/streamlining the manuscript. For example, removing figure 1 and table 3 may improve clarity.

Figure 1 has been removed from the paper and table 3 has been moved to the supplementary data.

2. Line 193: can you provide more details about "indeterminate results"? What caused these results? Were they due to failed internal control or failed RNase P assay? additionally, please clarify whether these indeterminate results occurred in self-collected swabs, health-care provider collected swabs or both. This information is important especially if RNase P assay was used to assess sample quality.

Indeterminate results occur in both swab types. Generally, the cause of indeterminate results is due to weak viral loads, detection of only a single target gene, or potential amplification of similar genes of other Coronavirus strains that share similar genes with SARS-CoV-2.

3. Are there significant difference between 2 Prongs arm or 1 prong arm? Please consider presenting these groups separately in tables 1 and 2.

As the sensitivity and specificity in Table 1 demonstrates consistency between the 1 prong and 2 prong arms of the study, a deeper analysis has not been performed. The main difference between the two is the average Ct value of the discordant pairs, with 2 prongs having an average of 35-36.3 and 1 prong having an average of 33.3-34. However, we did not do a statistical analysis between the two arms as they are not paired, we did not collect both a single prong, and a double prong from the same participant.

4. Please consider reformatting table 2. the layout of table 2 makes it difficult to follow.

The tables have been standardised to make them easier to follow.

5. Table 5: a 2 by 2 table may present the data more clearly. Please consider revising.

As per another reviewer's feedback, we have moved this table to the supplementary as the limited data allows only for speculation.

6. Line 142: please clarify the time interval between self-collection and the time when the swabs were picked up by the study staff. Prolonged storage of swabs in the refrigerator may affect the integrity of viral RNA and impact results.

All samples were delivered to the laboratory refrigerated within 72hrs of collection, as per the manufacturers IFU. However, the exact number of hours from collection to storage differs for samples as staff were performing multiple home visits each day and returning to the lab between each visit was not logistically possible.

7. Line 174: Ct values should be interpreted with caution as they are only a surrogate measure of viral load in a sample. Other factors also influence the Ct values. Please re-phrase this sentence.

We agree and have clarified the wording of this line.

8. There are grammatical errors throughout the manuscript. please revise

We have proofread and corrected errors throughout the text.

Re: Spectrum01711-25R1 (Diagnostic accuracy of self-collected anterior nasal swabs for SARS-CoV-2 rt-qPCR testing)

Dear Dr. Taylor Corocher:

Your manuscript has been accepted, and I am forwarding it to the ASM production staff for publication. Your paper will first be checked to make sure all elements meet the technical requirements. ASM staff will contact you if anything needs to be revised before copyediting and production can begin. Otherwise, you will be notified when your proofs are ready to be viewed.

Sincerely,
Wendy Szymczak
Editor
Microbiology Spectrum